# Depolymerization of SUMO chains induces slender to stumpy differentiation in *T. brucei* bloodstream parasites

**Paula Ana Iribarren**[1☉], **Lucía Ayelén Di Marzio**[1☉], **María Agustina Berazategui**[1¤a], **Andreu Saura**[2¤b], **Lorena Coria**[1], **Juliana Cassataro**[1], **Federico Rojas**[3], **Miguel Navarro**[2]*, **Vanina Eder Alvarez**[1]*

**1** Instituto de Investigaciones Biotecnológicas "Dr. Rodolfo Ugalde"–IIBIO (UNSAM-CONICET), San Martin, Buenos Aires, Argentina, **2** Instituto de Parasitología y Biomedicina "López-Neyra", CSIC (IPBLN-CSIC), Granada, Spain, **3** Biomedical and Life Sciences, Faculty of Health and Medicine, Lancaster University, Lancaster, United Kingdom

☉ These authors contributed equally to this work.
¤a Current address: Department of Life Sciences, Sir Alexander Fleming Building, Imperial College London, London, UK
¤b Current address: Life Science Research Centre, Faculty of Science, University of Ostrava, Ostrava, Czech Republic
* miguel.navarro@csic.es (MN); valvarez@iib.unsam.edu.ar (VEA)

**Data Availability Statement:** The RNA-seq data generated and analyzed during the current study

## Abstract

*Trypanosoma brucei* are protozoan parasites that cause sleeping sickness in humans and nagana in cattle. Inside the mammalian host, a quorum sensing-like mechanism coordinates its differentiation from a slender replicative form into a quiescent stumpy form, limiting growth and activating metabolic pathways that are beneficial to the parasite in the insect host. The post-translational modification of proteins with the Small Ubiquitin-like MOdifier (SUMO) enables dynamic regulation of cellular metabolism. SUMO can be conjugated to its targets as a monomer but can also form oligomeric chains. Here, we have investigated the role of SUMO chains in *T. brucei* by abolishing the ability of SUMO to polymerize. We have found that parasites able to conjugate only SUMO monomers are primed for differentiation. This was demonstrated for monomorphic lines that are normally unable to produce stumpy forms in response to quorum sensing signaling in mice, and also for pleomorphic cell lines in which stumpy cells were observed at unusually low parasitemia levels. SUMO chain mutants showed a stumpy compatible transcriptional profile and better competence to differentiate into procyclics. Our study indicates that SUMO depolymerization may represent a coordinated signal triggered during stumpy activation program.

## Author summary

SUMOylation is a reversible posttranslational modification found across eukaryotic organisms. It involves attaching a small ubiquitin-like modifier (SUMO) to specific lysine residues within proteins. This modification can happen at one lysine (monoSUMOylation), multiple lysines (multiSUMOylation), or through lysine-linked SUMO chains (polySUMOylation). By altering the interaction surface of proteins, SUMOylation acts

are available in the GEO repository under accession number GSE261736.

**Funding:** This work was supported by grants from the National Agency for Promotion of Scientific and Technological Research, from the Argentinian Ministry of Science and Technology (ANPCyT, MinCyT), grant PICT 2019-029000 to VEA and PICT 2017-0140 to PAI and grants from the Spanish Ministerio de Ciencia, Innovación y Universidades (RTI2018-098834-B-I00) (MINCIU-FEDER) and Redes y Centros de Investigación Cooperativa (RICET, https://www.ricet.es/) (M. N., RD16/0027/0019). The funders had no role in study design, data collection and analysis, decision to publish, or preparation of the manuscript.

**Competing interests:** The authors have declared that no competing interests exist.

like a switch, regulating their functions. This process is crucial for various cellular pathways, especially those involved in chromatin organization and function, such as transcription, replication, DNA damage repair, nucleocytoplasmic transport and chromosome segregation. SUMO chains, in particular, play a significant role during cellular stress.

In our study, we explored the role of SUMO chains in *Trypanosoma brucei*, an early branching pathogenic eukaryote.

We discovered that parasites capable of attaching only SUMO monomers are primed for differentiation from the slender replicative form to the stumpy non-replicative form. Our findings suggest that polySUMOylation of specific proteins helps maintain the replicative form of *T. brucei*, while reducing SUMO chain complexity may trigger the stress-induced transition to dormancy. This unveils a new function of polySUMOylation in controlling dormancy and, consequently, pathogenicity.

## Introduction

Pathogenic microorganisms have developed diverse strategies to colonize, multiply and survive in their hosts. *Trypanosoma brucei* spp, is a protozoan parasite and the causative agent of sleeping sickness in humans and nagana in cattle [1]. After infecting a mammalian host, the parasites invade and colonize multiple host tissues, such as the blood, the lymph, the skin, the adipose tissue and eventually the cerebrospinal fluid [2]. One of the main mechanisms to produce long-term infections is the antigenic variation of its major cell surface protein to evade the adaptive immune response of the host [3]. Another important feature is that the parasite can take control of its own replication to limit host demand, which would otherwise be lethal. This quorum sensing (QS) mechanism can initiate differentiation from a proliferative (slender forms, SL) to a quiescent stage (stumpy forms, ST). This is important not only to limit parasitemia, but also to spread the infection, as differentiated cells are pre adapted for survival in a completely different metabolic niche, such as the midgut of the tsetse fly vector [4]. The importance of a density-induced differentiation program becomes clear when comparing the course of infection of laboratory-adapted (monomorphic) and naturally occurring (pleomorphic) parasites. Monomorphic parasites have been selected *in vitro* as actively dividing cells but are unable to control their population size *in vivo*. Therefore, infection with monomorphic parasites inevitably ends in the death of the animal. In contrast, pleomorphic parasites have a density-induced differentiation pathway that allows the transition to an arrested form and can therefore produce persistent infections.

Several components of this QS differentiation pathway have been identified [5], including the stumpy inducing factor (SIF) and its transporter [6,7]. Very recently, a molecular mechanism involving a specific long noncoding RNA (lncRNA) that promotes the stability or translation of stumpy-inducing mRNAs has also been described [8]. Downstream signalling may involve cascades through the activity of the kinases TOR and AMPK [9], eventually leading to G1/G0 arrest, mitochondrial activation, biochemical changes, and morphological transformation. However, the full elucidation of the intracellular response is far from complete. Phosphorylation, like many other reversible post-translational modifications (PTMs) enables rapid modulation of protein function in response to environmental conditions. In this work, we have uncovered a link between another PTM, namely SUMOylation, and the differentiation process that occurs in the mammalian host.

SUMOylation is a PTM conserved in eukaryotic organisms involving the covalent attachment of the Small Ubiquitin-like MOdifier (SUMO) to internal lysine residues within target

proteins [10]. This modifier usually alters the interaction surface of its substrates, controlling the biological activity, stability, or subcellular localization, among other possible outputs. SUMOylation is essential in *T. brucei* [11,12] and regulates many important biological processes, as inferred by the proteomic analysis of its target proteins [13]. Interestingly, in *T. brucei* bloodstream forms (BSF) SUMO is enriched in a particular region of the nucleus, colocalizing with the RNA polymerase I at the expression-site body (ESB) within the active variant surface glycoprotein (VSG) expression site (VSG-ES). This highly SUMOylated focus (HSF) creates a permissive environment for VSG transcription [14].

Like ubiquitin, SUMO can modify substrates covalently as a monomer or by forming different types of chains through internal lysine residues within the N-terminus of SUMO, adding a new level of versatility and complexity to this dynamic PTM. SUMO chains have been described in many eukaryotic organisms. For example, they are involved in mitosis [15] or regulate the formation of promyelocytic leukemia nuclear bodies [16] and the synaptonemal complex [17]. We have shown that *Tb*SUMO is able to form polymeric chains via its K27 and that these structures are important for the assembly of nuclear foci, suggesting a regulatory role of polySUMOylation in chromatin organization in procyclic forms (PF) [18, 19].

In this work, we have investigated the role of SUMO chains in BSF using transgenic monomorphic cell lines that are unable to assemble them. We found that SUMO chain mutants can establish persistent infections in mice, in contrast to the virulent monomorphic parental strain. This behavior is related to a "stumpy-like" characteristic of the parasites, as the mutants exhibit some typical stumpy markers and increase differentiation kinetics from BSF to PF induced by *cis*-aconitate. Thus, in contrast to the "blind to the signal" status of the parental monomorphic cell line, the absence of SUMO chains renders BSF sensitive to population size. To confirm our results, we generated a similar mutant but using pleomorphic cells able to differentiate naturally. We found that the parasites are arrested *in vivo* at lower cell densities, exhibiting a typical stumpy morphology, and expressing the stumpy-specific marker PAD1. We propose that SUMO chain dynamics influence the ability to differentiate from SL to ST forms, being poly-SUMOylation a SL retainer signal that can be relieved after its depolymerization.

## Results

### SUMO chain mutant monomorphic parasites grow normally *in vitro* but have a reduced virulence in mice

To investigate the role of SUMO chains in the monomorphic BSF of *T. brucei*, we generated a mutant cell line in which all internal lysine residues in SUMO were replaced by arginine. Since SUMO is essential in *T. brucei*, we first investigated whether the absence of the chains could have a detrimental effect. SUMO*all*KR can be processed and conjugated as a monomer to target proteins at single or multiple lysine residues, resulting in mono- or multiSUMOylated adducts (*Tb*SUMO*all*KR, Fig 1A) that show a reduced intensity of the SUMOylation pattern in Western blot (Fig 1B), but without showing a growth phenotype in culture (Fig 1C). SUMOylation of chromatin at the active VSG expression site is a key feature of BSF [14]. Typical SUMO labeling analyzed by indirect immunofluorescence (IF) microscopy consists of a diffuse nuclear pattern and a highly SUMOylated focus (HSF) in ~65% of 1K 1N cells. In contrast, in the SUMO chain mutants, the signal was detected in both the nucleus and cytoplasm, with a more diffuse nuclear distribution (less than 40% of cells had an HSF) and a punctate signal in the cytoplasm, most likely due to the unconjugated fraction of SUMO (Fig 1D). However, this difference has no effect on VSG mRNA or protein levels (S1 Fig).

Next, we investigated the impact of SUMO chains *in vivo* using a mouse model of infection (Fig 2). Inoculation of wild-type (WT) parasites was lethal to 100% of mice within 6 days, as

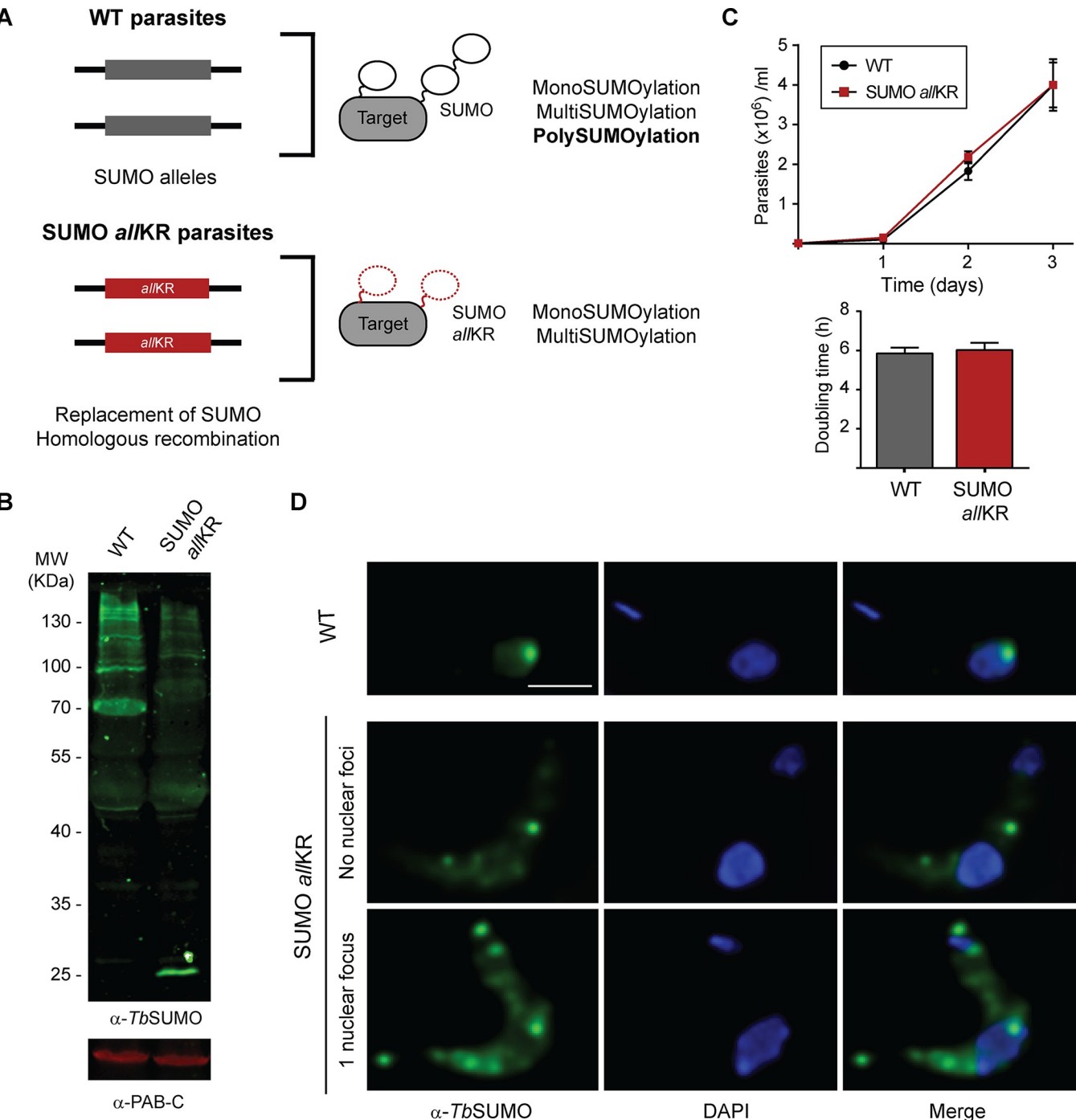

**Fig 1. Generation of SUMO chain mutant monomorphic BSF parasites.** (A) Schematic representation of the generation of *Tb*SUMO *all*KR parasites. (B) Conjugating ability of WT *Tb*SUMO (WT) and SUMO *all*KR parasites. Parasites were boiled in Laemmli's sample buffer immediately after harvesting. Proteins were separated by electrophoresis using a 10% SDS-poliacrylamide gel ($3\times10^7$ cells/lane). SUMO conjugates were analyzed by Western blot using anti-*Tb*SUMO antibodies and anti-PAB-C antibodies as loading control. (C) Growth curves for SUMO *all*KR and wild type (WT) parasites. WT and transgenic parasites were cultured up to one month without observing significant differences in growth rate. Doubling time was calculated by daily subculture back to $1 \times 10^5$/ml to maintain log-phase growth (*n* = 3). (D) Immunofluorescence (IF) analysis of WT and SUMO *all*KR BSF parasites. Nuclear and kinetoplast DNA were visualized by DAPI staining (blue). Representative images of anti-*Tb*SUMO (green) and anti-*Tb*SUMO-DAPI merged images are shown. For WT cells 63% of the nuclei showed a single HSF (*n* = 261) and for SUMO *all*KR cells this percentage was 37% (*n* = 318). Scale bar 5 μm.

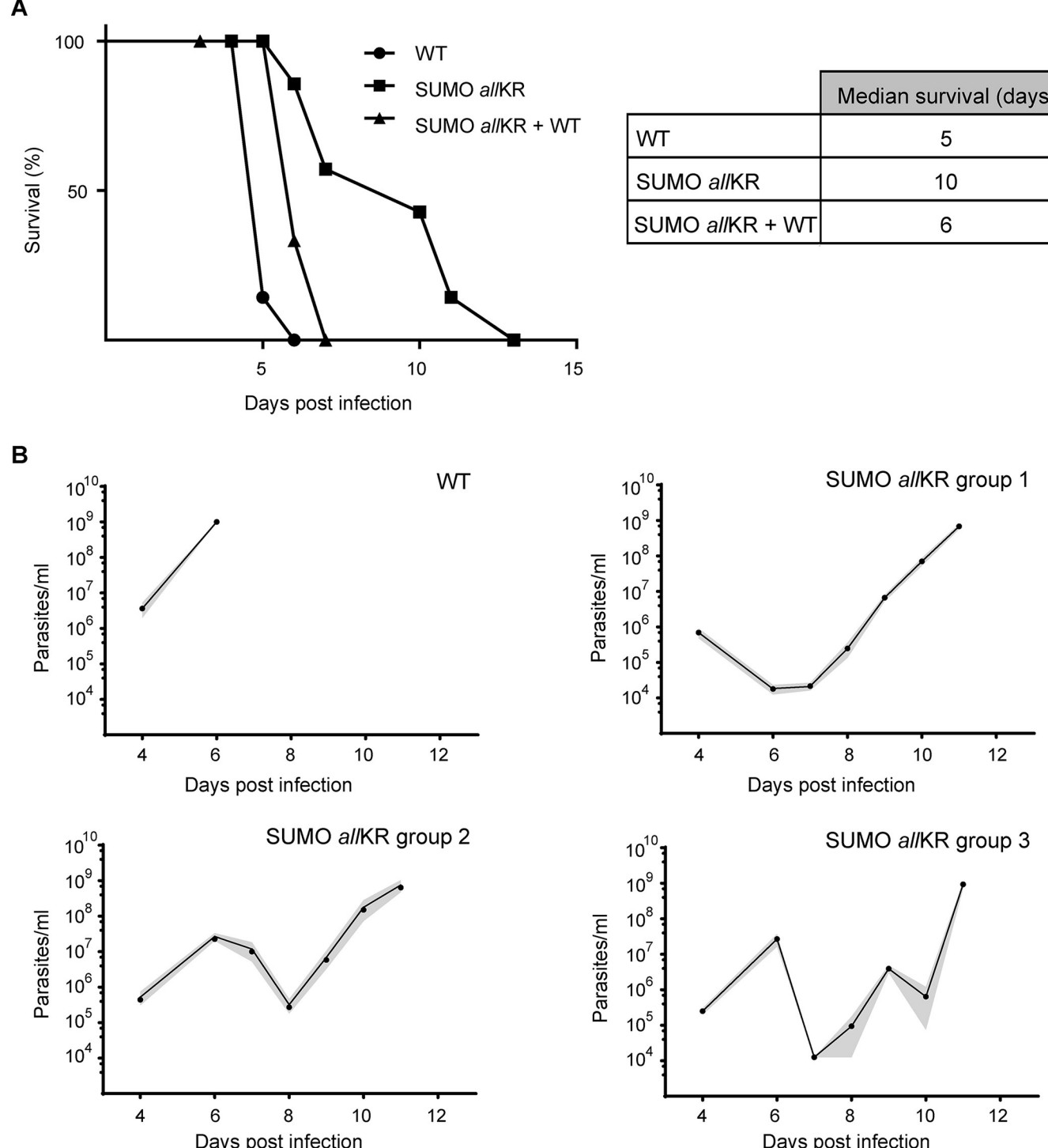

**Fig 2. Mice infections with SUMO chain mutant monomorphic BSF parasites. (A)** BALB/c mice were infected intraperitoneally with 5000 WT ($n = 7$, circles), SUMO *all*KR ($n = 7$, squares) or chain mutant parasites complemented with a WT copy of SUMO (SUMO *all*KR+WT; $n = 3$, triangles). Survival was monitored daily and is shown by a Kaplan-Meier curve. **(B)** Time course of parasitemia in mice infected with WT or SUMO *all*KR parasites. Animals were grouped according to their different parasitemia profile. Results are shown as the means (circles) and their corresponding SEM (gray shaded area). WT $n = 6$; SUMO *all*KR group 1 $n = 7$, group 2 $n = 7$, group 3 $n = 2$.

expected for a monomorphic cell line. In contrast, infection with SUMO chain mutant parasites resulted in significantly prolonged survival: 86% at day 6 post-infection (dpi) and 43% after 10 dpi. This phenotype was reversed when the mutant was complemented with a WT copy of SUMO, such that all mice died after 7 dpi, like animals infected with SUMO chain competent parasites (Fig 2A). To investigate this further, we monitored parasitemia over 11 days. In monomorphic WT-BSF, parasitemia was first detected at 4 dpi and increased 100-fold daily thereafter (n = 6). All mice developed a single wave of unremitting parasitemia with a terminal outcome at day 6 (Fig 2B, panel WT and S2 Fig). In contrast, mice infected with SUMO chain mutants (n = 17) showed two or three waves of parasitemia and in one case even cleared the infection (Fig 2B, panel *all*KR and S2 Fig).

## Monomorphic SUMO chain mutant parasites display stumpy-like characteristics

After ruling out possible differences in the anti-VSG IgM antibody response (S3 Fig) and in the ability of parasites to internalize and degrade anti-VSG antibodies (S4 Fig), we hypothesized that the absence of polySUMOylation might restore, at least in part, the original ability of the SL form to differentiate to ST in monomorphic parasites. At the peak of parasitemia, we observed cells that could not be recognized as genuine stumpy parasites based on their morphology or PAD1 expression (S5 Fig) [20]. However, we found a higher proportion of 1K 1N cells (Fig 3A) and increased mRNA levels for the characteristic stumpy markers PAD1 and PAD2 compared to WT cells (Fig 3B). To better characterize this mutant, we performed RNA-Seq on parasites isolated from the blood of infected mice at the peak of parasitemia. Analysis of the global expression profile revealed that 124 genes were differentially regulated between WT and SUMO chain mutants (Fig 3C and S1 Table). Of the 124 genes, 87 were upregulated while 37 were downregulated. Among the upregulated transcripts, we found 6 noncoding RNA in the top 10 and Tb927.10.12080, a reported target of the long non-coding RNA *grumpy* whose overexpression was associated with stumpy development [8]. In addition, PAD1 and 12 transcripts that are highly expressed in stumpy forms were also upregulated in these cells [21,22].

To further investigate whether SUMO chain mutants are primed for differentiation to stumpy forms, we challenged the parasites with high concentrations of *cis*-aconitate (CA) at low temperature to stimulate differentiation to PF, which is thought to occur via a stumpy-like intermediate stage [23,24]. During this transformation, the VSG coat is released and replaced by a different coat consisting of invariant GPEET and EP procyclins. In pleomorphic cell lines, which contain short stumpy cells, this process is synchronous and takes only 12 h. In monomorphic cells, however, the process is not synchronous and takes 36–48 h. To assess the ability of monomorphic, SUMO-chain competent or SUMO-chain mutant parasites to differentiate into PF, we monitored coat switching by IF analysis (Fig 3D). As expected, the transformation of SUMO chain-competent cells occurred with slow kinetics. After 24 h of CA treatment about 60% of the cells still expressed VSG, and this number decreased to 28% 24 h later. In contrast, the SUMO chain mutants showed significantly higher differentiation rates. Within 24 h, ∼90% of the cells were already EP-positive and VSG-negative, and transformation was almost complete after 48 h. Interestingly, these differences were abolished when parasites were pretreated with the cAMP analogue 8-pCPT-cAMP [8-(4-Chlorophenylthio) adenosine 3′, 5′-cyclic monophosphate] (a compound that can induce differentiation from slender to stumpy [25]) 24 h before the temperature shift and addition of CA (Fig 3E).

Overall, these results suggest that the absence of polySUMOylation renders monomorphic bloodstream parasites more sensitive to CA-triggered differentiation and that the underlying mechanism is upstream of AMP signalling.

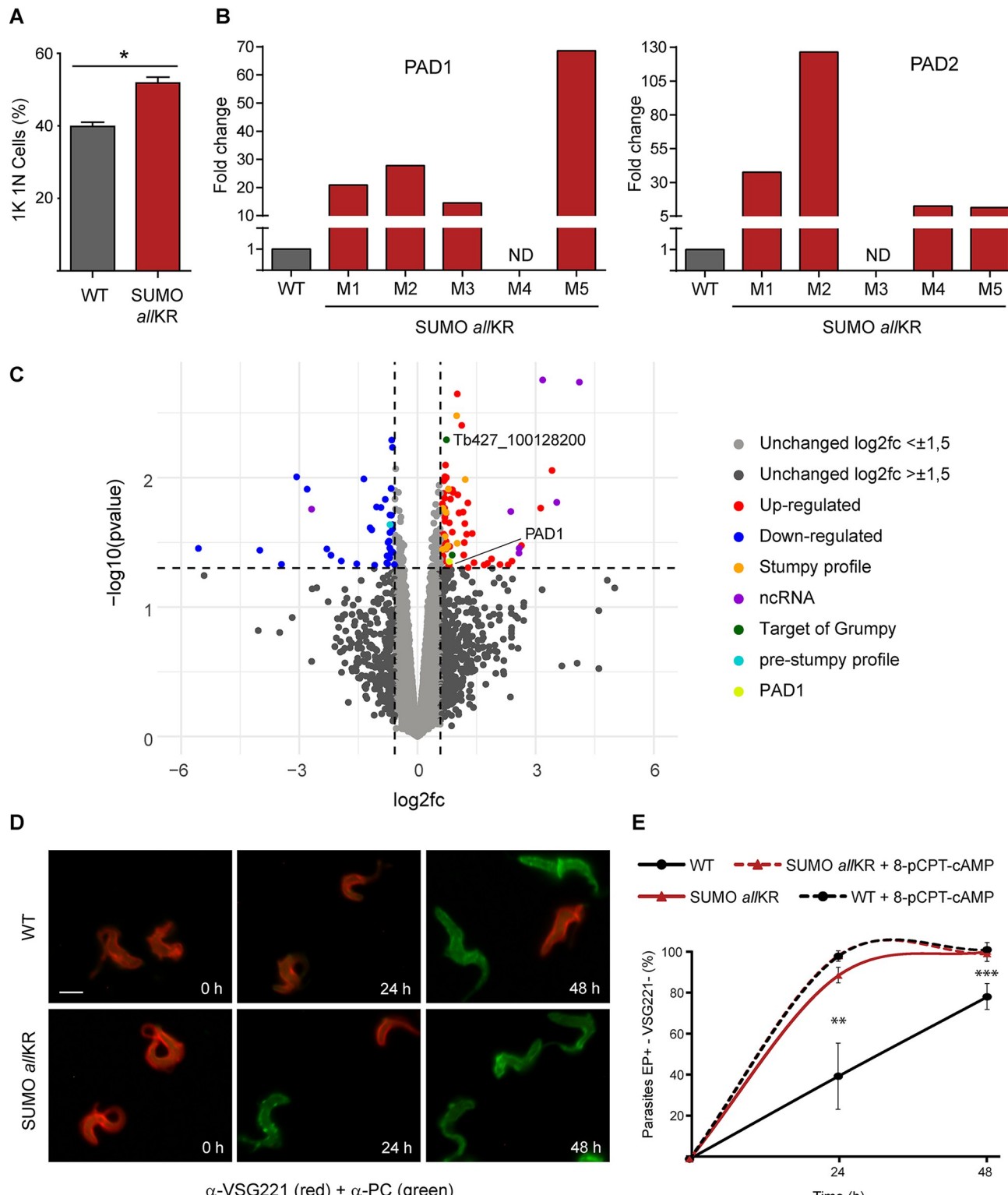

**Fig 3. Analysis of stumpy markers in monomorphic SUMO chain mutants. (A)** Blood smears from the first peak of parasitemia (5 dpi) were stained with DAPI for nucleus and kinetoplast configuration analysis. **(B)** Quantification of RNA transcript levels of the characteristic stumpy markers PAD1 and PAD2 was performed by qRT-PCR from RNA samples derived from mice infections (M1-M5 indicate mouse 1- mouse 5). WT and SUMO *all*KR parasites were isolated at the first peak of the parasitemia (at 5 dpi). Samples were normalized against 7SL. **(C)** Volcano plot showing the differential expression of genes analysis in SUMO *all*KR and WT parasites. Most significant up- and down-regulated genes are highlighted ([log2fc] ≥ 0.58,

pValue ≤ 0.05). Samples were collected on 5 dpi. **(D)** *In vitro* BSF to PF differentiation induced by CA was analyzed in SUMO *all*KR and WT parasites. Differentiation to the procyclic form was triggered in SDM-79 medium at 28°C using 6 mM CA. The differentiation process was evaluated following changes in the expression of stage-specific surface markers (VSG221 and procyclin) for 48 h by indirect immunofluorescence. Representative images of anti-VSG221 (red)-anti-EP (green) merged images are shown. One out of five representative experiments is shown. Scale bar 5 μm. **(E)** WT + 8-pCPT-cAMP and SUMO *all*KR + 8-pCPT-cAMP parasites were pre-treated with the cAMP analogue 8-pCPT-cAMP [8-(4-Chlorophenylthio) adenosine 3′,5′-cyclic monophosphate] during 24 h before shifting the temperature and adding CA. Results are shown as means (SEM, *n* = 5). Statistical test for (A) and (E): two-tailed paired t test (\*P < 0.01; \*\*P < 0.005, \*\*\*P < 0.0001). At least 100 cells were scored in each timepoint.

## SUMO chain mutant pleomorphic parasites are primed for stumpy differentiation

To investigate in detail the role of SUMO chain depolymerization in ST formation, we generated a similar SUMO chain mutant, but in differentiation-competent pleomorphic parasites. When the growth profile was examined (Fig 4A), a small effect on doubling time was observed when compared to WT parasites (Fig 4B). This growth phenotype is associated with an accumulation of non-dividing cells, as shown by the increase in the number of cells with 1K 1N configuration (Fig 4C). To analyze the formation of ST *in vitro*, we exposed the parasites to an increased local concentration of SIF by culturing them on HMI-9 agarose plates for 4 days followed by immunodetection of the stumpy marker protein PAD1. As shown in Fig 4D and 4E, chain mutant parasites had a significantly higher proportion of cells expressing PAD1 (74% ± 2%), while only 35% ± 6% of WT cells showed expression of PAD1 at the same time point. Furthermore, while PAD1 was already present on the surface of the chain mutant parasites, it was

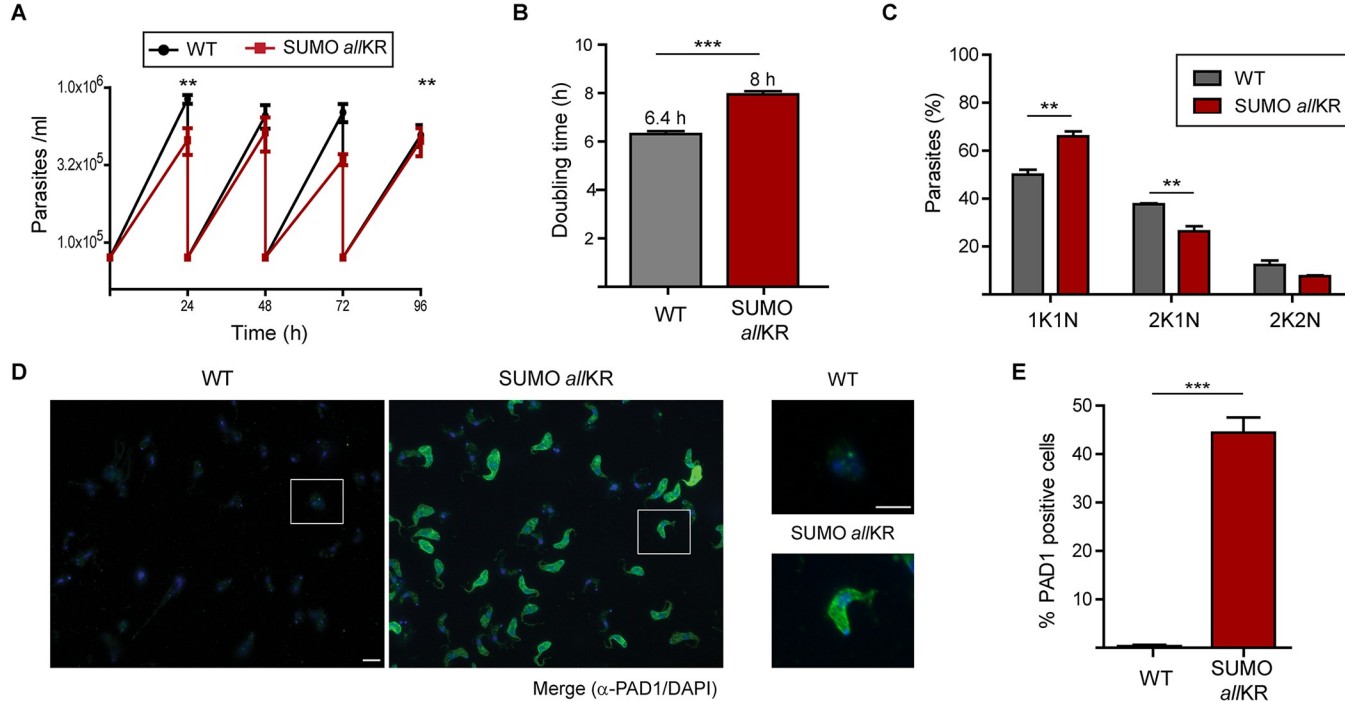

**Fig 4. Generation of SUMO chain mutant pleomorphic parasites. (A)** Growth curves of pleomorphic parasites (*n* = 3). WT or SUMO *all*KR parasites were cultured *in vitro* under regular conditions in HMI-9. **(B)** Doubling time was calculated by daily subculture back to 1×10⁵/ml to maintain log-phase growth. **(C)** Pleomorphic parasites in exponential growth phase were fixed and stained with DAPI for nucleus and kinetoplast configuration analysis (*n* = 3). At least 100 cells were scored in each replicate. **(D)** Stumpy formation in HMI-9 agarose plates. WT or SUMO *all*KR parasites were plated onto semi solid agarose plates. After 4 days, parasites were harvested, fixed and stained with DAPI (blue) and anti-PAD1 antibodies (green) (*n* = 3). Representative images and insets are shown. Scale bar 5 μm. **(E)** Quantification of parasites from (D). At least 100 cells were scored in each experiment. Statistical test: two-tailed paired t test (\*\*P < 0.01; \*\*\*P < 0.001).

observed in vesicles and in the flagellar pocket in the WT cells, most likely indicating traffic to the membrane. Finally, the chain mutants showed an increased rate of transformation to PF when exposed to the differentiation signal CA (S6 Fig).

Having confirmed that the absence of the SUMO chains renders the parasite more susceptible to QS signalling *in vitro*, we monitored the consequences in mouse infections. Figs 5A and S7 show that infection with SUMO*all*KR pleomorphic parasites resulted in consistently reduced parasitemia and prolonged host survival (Fig 5B). This pattern of infection is driven by a premature growth arrest (Fig 5C) as >95% of SUMO *all*KR parasites had a 1K1N configuration at the first peak, when parasitemia was only $10^6$ parasites/ml. Even at these low densities, we observed that the parasites were morphologically stumpy and displayed the stumpy-specific marker protein PAD1 on their surface (Fig 5D and 5E), which is in contrast to the pleomorphic WT cells. Our results thus clearly show that the absence of SUMO chains facilitates stumpy development during infections.

## Discussion

The results obtained in this work show that SUMO chains in *T. brucei* BSF represent an inhibitory signal for differentiation from slender to stumpy and that disassembly of these chains in the context of infection may promote persistence of the parasite in the host. Our conclusion is based on several experimental findings: 1) attenuated virulence for monomorphic and pleomorphic parasites; 2) stumpy-like transcriptome profile in monomorphic parasites and premature PAD1 protein expression in pleomorphic parasites; and 3) the accelerated rate of differentiation to PF following *cis*-aconitate exposure in monomorphic and pleomorphic parasites (Fig 6).

In this study, we mimicked the monoSUMOylation status of target lysine residues and thus abolished the ability of SUMO to polymerize. This was achieved by substituting the native SUMO alleles with a variant where all lysine residues were transformed into arginine (SUMO *all*KR), maintaining the overall charge while hindering subsequent internal modification of SUMO through SUMO. We have chosen this strategy instead of specific mutation of the lysine residue at position 27, as previous reports had shown that in the absence of the main acceptor lysine residue, some other lysines (otherwise cryptics) can be modified, in some cases with the assistance of SUMO ligases [26–32]. This phenomenon has also been observed with ubiquitin [33,34]. In view of this, and because we wanted to analyze the effects of chain deficiency, we decided to replace all lysine residues to ensure that no chains were present.

SUMO chains are not essential for the axenic growth of BSF, at least under our experimental conditions using routine HMI-9 culture medium. Remarkably, the SUMO chain mutant parasites express average levels of VSG, suggesting that monoSUMOylation of specific factors adequately recruits PolI to the active expression site. The most surprising phenotype for monomorphic or pleomorphic mutant cell lines was detected *in vivo*. We consistently observed that the abrogation of SUMO chains resulted in reduced virulence and increased host survival in *all*KR parasites. In monomorphic parasites, mice infected with SUMO chain mutants showed relapsing and remitting waves of parasitemia, reaching densities between $10^8$–$10^9$/ml and clearing the day after. This behaviour is completely unusual for a monomorphic cell line, as these developmentally incompetent parasites cannot control parasitemia by differentiating into quiescent stumpy cells and kill the host within the first few days of infection, as seen with parental WT parasites. Pleomorphic strains, on the other hand, can complete the natural life cycle of the parasite and show persistent infections with undulating parasitemia in animal models [4,35,36]. Thus, it appears that the abrogation of SUMO chains has attenuated the virulence of monomorphic strains *in vivo*, prolonging mouse survival and decreasing parasitemia. To our knowledge, this characteristic growth pattern has only been reported for

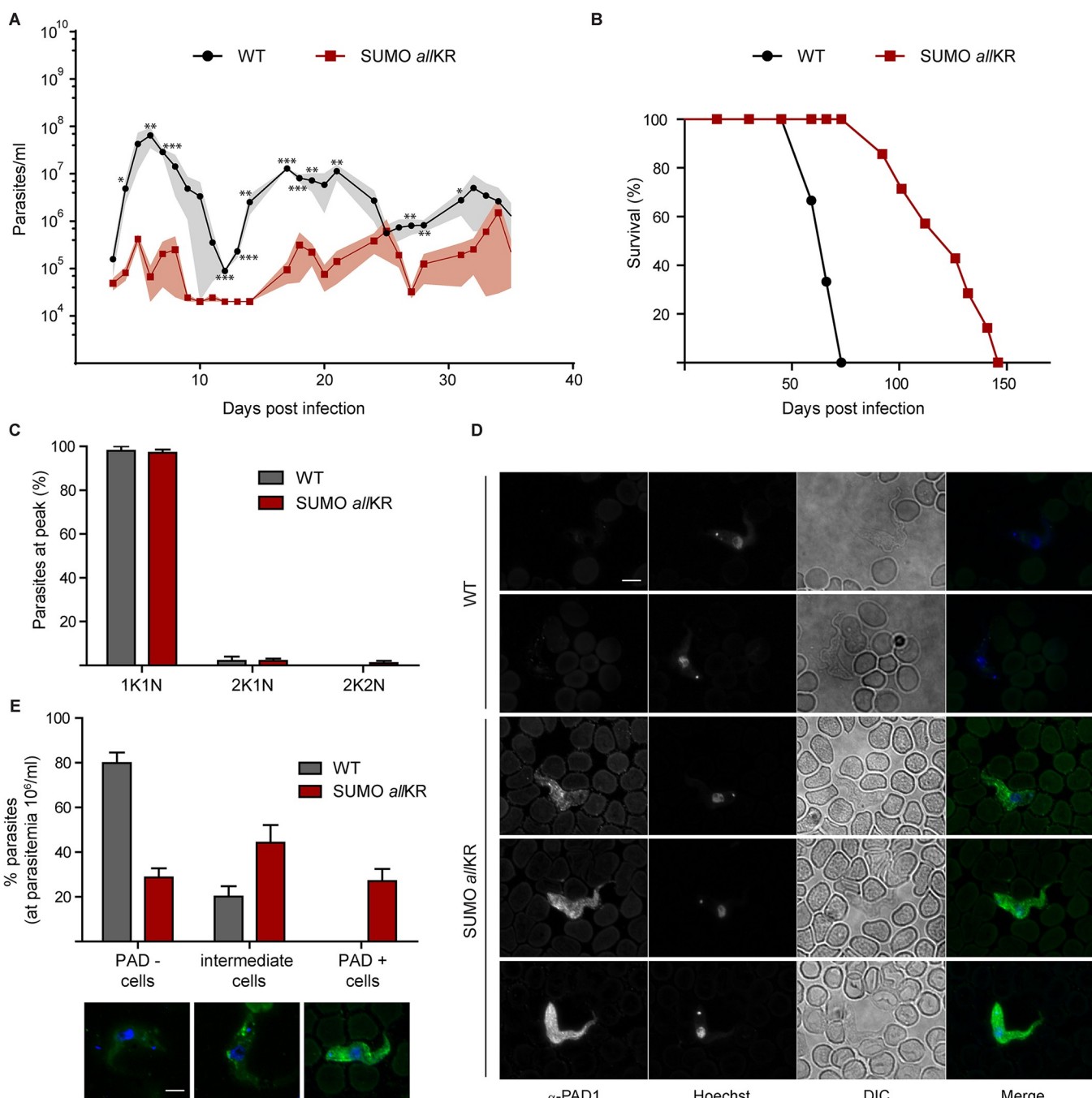

**Fig 5. Mice infections with SUMO chain mutant pleomorphic parasites.** BALB/c mice were infected intraperitoneally with 5000 WT or SUMO *all*KR parasites. (**A**) Time course of parasitemia in mice infected with WT (*n* = 3, black circles) or SUMO *all*KR parasites (*n* = 7, red squares). Results are shown as the means and their corresponding SEM (gray shaded area). (**B**) Mice survival was monitored daily and is shown by a Kaplan-Meier curve for infections with WT (*n* = 3, black circles) and SUMO *all*KR parasites (*n* = 7, red squares). (**C**) Nucleus and kinetoplast configuration was analyzed in methanol-fixed blood smears stained with Hoechst. (**D**) Representative images of methanol-fixed blood smears of infected mice prepared and stained with anti PAD1 antibodies (green) and Hoechst (blue). (**E**) Quantification of PAD1 stained parasites of (D). The bottom panel shows representative images of parasites expressing PAD1 in the surface (PAD1 positive), parasites that do not show any labelling (PAD1 negative) and parasites that express low levels of PAD1 mainly in vesicles (intermediate cells). For (C-E) parasites were collected from the first peak of parasitemia (at 6 dpi). Experiments were performed in triplicates (WT, *n* = 36; SUMO *all*KR, *n* = 62). Scale bar 5 μm. Statistical test: two-tailed paired t test (*P < 0.05; **P < 0.01; ***P < 0.001).

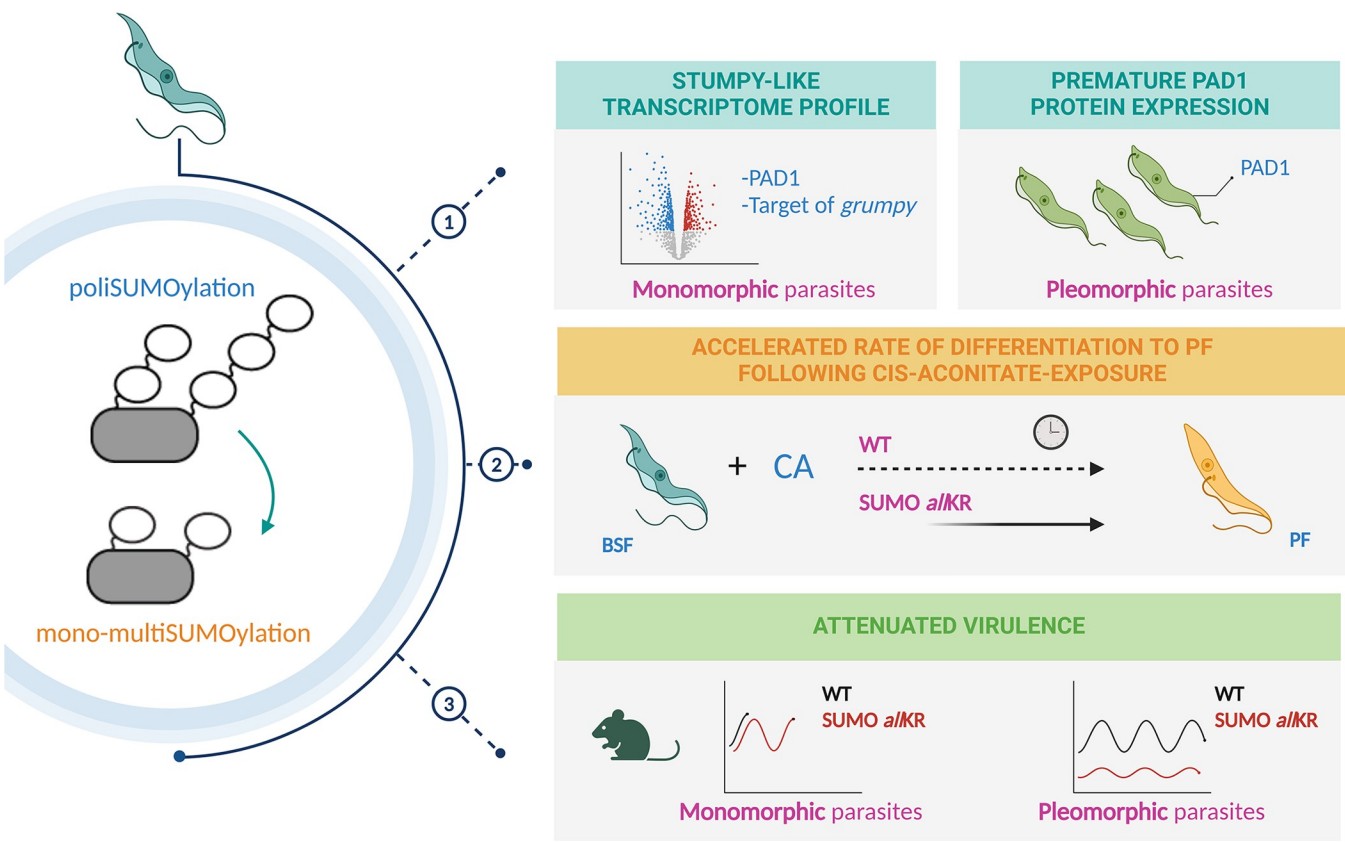

**Fig 6. Schematic diagram summarizing the main experimental findings.** Image was created with BioRender.com.

one other monomorphic strain, which was a null mutant for the calflagin gene [37], although the underlying mechanism has not been investigated further.

After discarding potential differences in the anti VSG IgM antibody response and in the ability of the parasites to internalize and degrade anti-VSG antibodies, we hypothesized that the altered course of infection might be the result of differentiation into stumpy-like cells. Indeed, the expression of the characteristic stumpy markers PAD1 and PAD2 was increased at the mRNA level (but not at the protein level) compared to control monomorphic WT parasites. These intermediate phenotypes, which have not yet completed the developmental process, have already been described. They express PAD1 mRNA prior to protein expression and final morphological transformation into mature stumpy cells [36, 38–43]. The analysis of the global transcriptional profile of the monomorphic chain mutants obtained by RNAseq shows an enrichment of transcripts previously identified as characteristic of stumpy forms [21, 22]. On the other hand, it was noticed that several noncoding RNAs are found among the most abundant transcripts. It was recently described that a lncRNA, which gives rise to a small nucleolar RNA called *grumpy*, binds to several mRNAs involved in the transition to stumpy forms and promotes their expression [8]. Interestingly, one of the mRNA targets of *grumpy* (Tb427_100128200), whose overexpression has been experimentally confirmed to trigger premature differentiation, is also upregulated in our chain mutants, supporting the notion that these parasites are committed to the differentiation pathway. Remarkably, this phenotype was only observed *in vivo*, while *in vitro* cultured SUMO *all*KR parasites were negative in all assays used to evaluate stumpy characteristics (morphology, mild acid resistance, mitochondrial activity, cell cycle status and expression of stumpy markers; data not shown) [20] except for

differentiation using CA stimulus. In this latter case, the SUMO chain mutants displayed accelerated kinetics of differentiation to PF compared to WT parasites by showing significantly increased replacement of the VSG coat by procyclin 24 h after induction. Furthermore, this difference was abolished by treatment with a cAMP analogue, suggesting that SUMO chain signalling may be upstream of the AMPK pathway [9]. Taken together, these results suggest that, in contrast to monomorphic WT strains, the absence of polySUMOylation renders parasites more susceptible to differentiation, while host-parasite interaction is required to trigger stumpy-like cells. In agreement with this, Rojas et al. have recently described the importance of the interplay between environmental cues and trypanosomes, as both host and parasite proteins, are required for quorum sensing signaling in developmentally competent cells [6].

We have confirmed our hypothesis by generating a similar mutant in a differentiation competent pleomorphic background. Mutant parasites grown on HMI-9 agarose plates showed all the typical characteristics of the stumpy form, including expression of the PAD1 protein at the cell surface, arrest at the G0-G1 phase of the cell cycle, and prompt differentiation to PF when incubated with CA. Furthermore, these mutants differentiated into stumpy forms *in vivo* at low densities (approximately $10^6$–$10^7$ parasites/ml), which was associated with lower parasitemia and longer survival of infected mice. Differentiation into stumpy forms is not exclusively regulated by cell density-dependent mechanisms and various stimuli and stressors can trigger the formation of stumpy cells. Furthermore, it has been hypothesized that *T. brucei* evolved mechanisms that merged stress responses with pathways controlling differentiation [44].

SUMO chains have been implicated in several processes. In particular, they have been found to play a crucial role when cells are under stress [45,46]. The best characterised example is proteotoxic stress, where they act as a signal for protein degradation and target polySUMOylated proteins for ubiquitination [47]. In addition, other studies [48,49] have demonstrated that the SUMO system is involved in transcriptional control. One study showed that SUMO chains are responsible for the activation of survival pathways during stress through the transcriptional derepression of stress-regulated genes [50]. To date, site-specific proteomic studies have allowed the identification of SUMO targets in PF and BSF parasites, many of which are associated with relevant nuclear processes such as DNA replication and repair, RNA metabolism, transcription and chromatin remodelling [13]. Considering that transcription and protein synthesis are downregulated during the transition from slender to stumpy forms [20], some of the reported SUMO-modified chromatin remodelers could be interesting candidates to investigate their polySUMOylation status with respect to differentiation. In mammalian cells, the dynamics of polySUMOylation depends on the activity of specific SUMO proteases (SENP6 and SENP7) and their regulation is crucial for the crosstalk between ubiquitin and SUMO [51]. Interestingly, the mRNA profile of the major SUMO-deconjugating enzyme in *T. brucei*, *Tb*SENP [18,52], was increased in the differentiation-competent AnTat1.1 cells during synchronous transformation and reached the highest levels in the stumpy forms derived from infections in mice ([53], Additional file 3b).

Based on our results, we propose that polySUMOylation of specific substrates is important to maintain *T. brucei* in a slender form while debranching of SUMO chains might reduce the threshold required for the stress activated differentiation program. We believe that the dynamics of the SUMO chains represent an additional level of control of this PTM and a novel layer that can be modulated to influence parasite differentiation.

## Materials and methods

### Ethics statement

Animal experiments were approved by the Committee on the Ethics of Animal Experiments of the Universidad Nacional de San Martin (CICUAE-UNSAM #11/17) and were carried out

according to the recommendations of the Guide for the Care and Use of Laboratory Animals of the National Institutes of Health and the guidelines laid down by the Committee for the Care and Use of Animals for Experimentation.

## Trypanosome culture

*Trypanosoma brucei* bloodstream form (BSF) "Single Marker" (SM) parasites (T7RNAP TETR NEO), pleomorphic BSF EATRO 1125 [54] and transfected cells were grown at 37°C and 5% $CO_2$ in HMI-9 media [55] (Life Technologies, Carlsbad, CA, USA) supplemented with 10% (vol/vol) heat-inactivated fetal calf serum (Natocor, Córdoba, Argentina) and appropriate antibiotics.

## Generation of SUMO chain mutant parasites

For SUMO *all*KR strain, we employed a synthetic construct (GenScript, Piscataway, NJ, USA) described previously [19]. This construction contained the coding sequence for 8 histidine residues, an hemagglutinin (HA) tag, the complete open reading frame (ORF) of *Tb*SUMO (Tb927.5.3210) with all its 8 lysine residues replaced by arginine residues, a 200 bp fragment of the 5´end of 3´untranslated region (UTR) of the gene, a *Xho*I restriction site and 250 bp of the 3´end of *Tb*SUMO 5´UTR. This was cloned into the endogenous locus tagging vector pEnT6 [56], with puromycin or hygromycin resistance marker cassette, to allow the sequential replacement of *Tb*SUMO alleles by homologous recombination through its UTRs. The same vector, but with WT *Tb*SUMO ORF [13] and blasticidin resistance marker cassette, was employed to reverse SUMO *all*KR phenotype. Vectors were linearized with *Xho*I (New England Biolabs, Ipswich, MA) and used to transfect BSF parasites based on the protocol described by the Cross laboratory (http://tryps.rockefeller.edu/). For the SUMO *all*KR pleomorphic strain, one allele of *Tb*SUMO was first replaced by a blasticidin resistance marker cassette. To achieve this, a pGEM-T Easy vector (Promega, Madison, WI, USA) with the blasticidin cassette flanked by a fragment of the 5' UTR and 3' UTR of the SUMO gene, to allow replacement by homologous recombination, was made. This vector was linearized with *Not*I (New England Biolabs, Ipswich, MA) and used to transfect BSF parasites. After confirming the correct replacement of the blasticidin cassette, this SUMO hemi KO line was transfected with the *Tb*SUMO *all*KR construct, as described for the monomorphic strain.

Log phase cells (2–3 x$10^6$ ml$^{-1}$) collected by centrifugation were resuspended in 90 µl of Tb-BSF buffer (90 mM $Na_2HPO_4$, 5 mM KCl, 0.15 mM $CaCl_2$, 50 mM HEPES, pH 7.3) and mixed with 10 µl of linearized DNA (5–15 µg) in a 0.2 cm electroporation cuvette (BTX, Harvard Apparatus, Holliston, MA, USA). Parasites were then subjected to one pulse using X-001 program in the Amaxa Nucleofector 2b (Lonza Cologne AG, Germany). Transfected cells were cloned after 6 h in 24-well dishes with the appropriate selective drugs (2.5 µg/ml of G-418; 0.1 µg/ml of puromycin; 5 µg/ml of hygromycin; 5 µg/ml of blasticidin (InvivoGen, San Diego, CA, USA)). The correct replacement of WT *Tb*SUMO alleles in SUMO *all*KR clones was confirmed by Polymerase Chain Reaction (PCR) using specific primers, as described previously [19]. *Tb*SUMO PCR products were sequenced (Macrogen, Seoul, Korea).

## Electrophoresis and immunoblotting

Parasites collected by centrifugation were resuspended in Laemmli sample buffer and boiled for 5 min. Protein extracts were resolved on SDS-PAGE (10% acrylamide) and transferred to a nitrocellulose Hybond ECL membrane (GE Healthcare, Pittsburgh, PA, USA) for probing with high-affinity rat monoclonal antibodies anti-HA (Roche, Basel, Switzerland) diluted 1:500, mouse anti-VSG221 diluted 1:500 and mouse anti-*Tb*SUMO diluted 1:500. Mouse

monoclonal anti α-tubulin clone B-5-1-2 (Sigma-Aldrich, St. Louis, MO, USA) and anti-PABP [57] were used as loading controls. Alexa Fluor 790 AffiniPure goat anti-mouse IgG (H+L) or Alexa Fluor 680 AffiniPure goat anti-rabbit IgG (H+L) secondary antibodies (Jackson Immunoresearch Laboratories, West Grove, PA, USA) diluted 1:25000 were detected using an Odyssey laser-scanning system and quantified with Image Studio software (LI-COR Biosciences, Lincoln, NE, USA). Antibody signals were analyzed as integrated intensities of regions defined around the blots of interest.

## Growth curves

Parasite growth was evaluated by counting cell numbers daily by quadruplicate in a Neubauer haemocytometer. For doubling time calculation, parasites were maintained on exponential growth by diluting cultures every day to a density of $1x10^5$ cells/ml.

## Indirect immunofluorescence

BSF cells were collected by centrifugation (1000 x g for 10 min), washed with TDB supplemented with glucose 20 mM and fixed with 4% paraformaldehyde (PFA) in PBS for 1 h. Parasites were allowed to bind to poly-L-lysine coated glass coverslips for 30 min and then incubated with 25 mM $NH_4Cl$ for 15 min. Permeabilization and blocking were performed with 3% bovine serum albumin (BSA), 0.5% saponin and 5% normal goat serum in PBS for 1 h. Mouse anti-*Tb*SUMO (1:500) [14], mouse anti-VSG221 (1:200), rabbit anti-VSG221 (1:500), mouse anti-EP FITC (1:500) (Cederlane Laboratories, Burlington, Canada) or anti-PAD1 [58] were used as primary antibodies. After washing with PBS, coverslips were incubated for 1 h with secondary antibodies diluted 1:1000 in 1% BSA:PBS (polyclonal goat anti-rabbit Alexa Fluor 568 or polyclonal goat anti-mouse Alexa Fluor 488 (Jackson)). Finally, coverslips were extensively washed and mounted using FluorSave reagent (Merck, Darmstadt, Germany). Nucleus and kinetoplast were visualized with 4,6-diamidino-2-phenylindole (DAPI) (Life Technologies). Samples were analyzed with an Eclipse 80i microscope (Nikon, Shinagawa, Japan). For SUMO visualization images were acquired as 3D z-stacks, deconvoluted and projected into 2D using a maximum intensity projection, as previously done in López Farfán et al., 2014 [14].

## Flow cytometry

Parasites were collected by centrifugation, washed in TDB- glucose 20 mM and fixed with 1% PFA for 30 min at 4°C. After washing with PBS, cells were incubated with mouse anti-VSG221 (1:200) or mouse anti-EP FITC (1:500) diluted in 1% BSA:PBS for 30 min with gentle stirring. Parasites were then washed with PBS and resuspended in secondary antibodies diluted in 1% BSA:PBS (polyclonal goat anti-mouse Alexa Fluor 488 (Jackson) diluted 1:200). Finally, cells were extensively washed with PBS and analyzed in a BD LSRFFortessa X-20 Cell Analyzer (BD, USA). Data analysis was performed using the FlowJo software (FlowJo LLC, Ashland, OR USA).

## *In vitro* differentiation

To promote differentiation to PF, BSF parasites were resuspended at a density of $2.5x10^6$ cells/ml in SDM-79 media (Life Technologies) with 6 mM cis-aconitate (Sigma-Aldrich) at 28°C. Samples were collected by centrifugation at different time points and analyzed by immunofluorescence.

For stumpy differentiation, pleomorphic BSF parasites from a mid-logarithmic growth phase culture were grown in semisolid HMI-9 agarose plates at low density ($1x10^5$-$1x10^6$ cells

per plate). After 4 days of incubation at 37˚C and 5% $CO_2$, the stumpy-enriched population was harvested from the agarose plates by washing with HMI-9 and collected by centrifugation [59].

## Mice infections

Female BALB/c mice were inoculated intraperitoneally with 5000 parasites obtained from axenic culture in exponential growth. Parasitemias were obtained by counting cell number in blood with 0,83%w/v ammonium chloride by quadruplicate in a Neubauer chamber under the light microscope (×400).

## RNA isolation and gene expression analysis by RT-qPCR

Total RNA was isolated from ~$5x10^7$ BSF parasites obtained from an axenic culture in exponential growth or blood of infected mice with Trizol Reagent, according to manufacturer´s instructions (Life Technologies). RNA integrity was assessed by electrophoresis and genomic DNA was eliminated with RQ1 DNAse (Promega, Madison, WI, USA) and subsequent chloroform extraction and ethanol precipitation. RNA was quantified by spectrophotometric assay with NanoDrop system. cDNA was obtained from 3 μg of total RNA using 200 U Superscript II reverse transcriptase (Life Technologies) and 200 ng of random primers in a 20 μl total volume. Reactions were incubated at 42˚C for 50 minutes and then at 70˚C for 15 minutes. Quantitative PCR assays were carried out in the 7500 Real Time PCR System from Applied Biosystems using SensiFAST SYBR Lo-ROX Kit (BioLine, London, UK) and primers previously described [9,60].

## ELISA assays

Microplates containing 96 wells (Thermo Scientific ImmunoPlates, MaxiSorp, Waltham, MA, USA) were coated overnight at 4˚C with 5 μg/well of VSG221 purified as described by Cross [61] in PBS pH 7.4. Plates were washed 2 times with TBS-Tween (50 mM Tris-HCl (pH 7.6), 150 mM NaCl, 0.05% (v/v) Tween) and then blocked with 200 μl/well of buffer 1 (5% (w/v) skimmed milk in TBS-Tween) at room temperature (RT) for 1 h. The plates were then incubated for 1 h at RT with sera from infected mice diluted 1:5 in buffer 1. After washing 4 times with TBS-Tween, 100 μl of secondary antibody diluted in buffer 1 (peroxidase-conjugated goat anti-mouse IgM antibodies (Sigma-Aldrich)) were added and incubated at RT for 1 h. After additional washings with TBS-Tween the reaction was developed with tetramethylbenzidine for 15 min (TMB, Sigma-Aldrich) and stopped with 0.2 M sulphuric acid. Finally, absorbance values were measured at 450 nm in a microplate absorbance reader (FilterMax F5 Multimode, Molecular Devices, Sunnyvale, CA, USA).

## RNA-seq analysis

Total RNA was isolated as previously described from parasites of the first peak of parasitemia, obtained by differential centrifugation. At least two independent biological samples for each cell line were sequenced as described by Saura et al. [62]. Genes transcripts from *Tb*SU-MO*all*KR versus WT parasites with [log2FC] $\geq$ 0.58 (log2 of Fold Change) and a pValue $\leq$ 0.05 were considered differentially expressed. Volcano plot was constructed using R with the package ggplot2 (https://ggplot2.tidyverse.org/). The RNA-seq data generated and analyzed during the current study are available in the GEO repository under accession number GSE261736.

## Supporting information

**S1 Fig. Analysis of VSG expression in SUMO chain competent *versus* SUMO chain mutant BSF parasites. (A)** Quantification of RNA transcript levels corresponding to the VSG221 gene expressed in WT and SUMO *all*KR parasites. Transcript levels were determined using qRT-PCR and normalized against 7SL and C1 (*n* = 3). **(B)** VSG221 expression levels were evaluated by Western blot analysis in total cell extracts with anti-VSG221 antibodies and tubulin as a loading control. Experiments were performed at least in triplicates and one representative image is shown. Quantification of band intensities was performed using ImageJ. **(C)** The density of the VSG coat in WT and SUMO *all*KR parasites was analyzed in fixed cells with anti-VSG221 antibodies and flow cytometry. Approximately 50000 events were captured. Representative flow cytometry histograms normalized to mode are shown. MFI: median fluorescence intensity.
(TIF)

**S2 Fig. Mice infections with SUMO chain mutant BSF parasites.** Time course of parasitemia in mice infected with **(A)** WT or **(B)** SUMO *all*KR parasites. Animals were grouped according to their different parasitemia profile. In animals infected with WT cells, parasitemia was first detected at 4 dpi and increased 100-fold daily thereafter. All mice developed a single wave of unremitting parasitemia with a terminal outcome at day 6. In the majority of the mice infected with SUMO chain mutants, parasites proliferated reaching the highest cell density approximately at 4–6 dpi after which the number of cells abruptly dropped at day 6–8 increasing again the day after. In general, all mice reached a maximum density of ~$5 \times 10^7$-$1 \times 10^8$ parasites/ml before dropping, while values around $5 \times 10^8$-$10^9$ parasites/ml led inexorably to death. Animal death before the end of the experiment is indicated with an asterisk. The mouse in SUMO *all*KR group 4 cleared the infection.
(TIF)

**S3 Fig. Anti-VSG IgM antibody response.** Detection of IgM antibodies against VSG221 in the serum of mice infected with WT or SUMO chain mutant (SUMO *all*KR) parasites at the first peak of parasitemia by ELISA. Serum samples were diluted 1:5, and absorbance values (Abs) were measured at 450 nm. C+: positive control; C-: negative control.
(TIF)

**S4 Fig. Degradation of anti-VSG IgG.** WT or SUMO chain mutant (SUMO *all*KR) parasites were first incubated with anti-VSG221 IgG at 4˚C in HMI-9, followed by an incubation at 37˚C for 5 and 10 minutes. **(A)** Parasites were fixed and stained with anti-mouse-Alexa Fluor 488 (green) and DAPI (blue). **(B)** Whole-cell extracts (corresponding to $1 \times 10^7$ cells) were obtained after the 10-minute incubation at 37˚C and boiled in Laemmli's sample buffer. Proteins were separated by SDS-PAGE and transferred to a nitrocellulose membrane, followed by immunoblotting with anti-mouse-Alexa Fluor 790.
(TIF)

**S5 Fig. Evaluation of stumpy parasites *in vivo*.** Representative images of methanol-fixed blood smears of infected mice on the peak of parasitemia (at 5 dpi) stained with **(A)** Giemsa; **(B)** anti PAD1 antibodies (green) and DAPI (blue). Stumpy cells from pleomorphic parasites are shown as positive control (arrowhead). Scale bar 10 μm.
(TIF)

**S6 Fig. CA-induced differentiation of pleomorphic parasites.** WT or SUMO *all*KR parasites were harvested from semisolid HMI-9 agarose plates and incubated with 6 mM cis-aconitate in SDM-79 at 28˚C. Parasites were collected by centrifugation after different time points, fixed

and stained with anti-EP FITC. Samples were analyzed by flow cytometry.
(TIF)

**S7 Fig. Mice infections with pleomorphic parasites.** Time course parasitemia in mice infected with **(A)** WT or **(B)** SUMO *all*KR parasites.
(TIF)

**S1 Table. Differential expression of genes analysis in SUMO *all*KR and WT parasites.**
(XLSX)

**S1 Data. Excel spreadsheet containing, in separate sheets, the underlying numerical data for Figs** 1C, 2A, 2B, 3A, 3B, 3E, 4A, 4B, 4C, 4E, 5A, 5B, 5C, 5E, S1A, S1B, S2A, S2B, S3, S7A and S7B**.**
(XLSX)

## Acknowledgments

We are grateful to G. Montagna and J. De Gaudenzi (IIBIO-UNSAM, Argentina) for helpful discussion and critical reading of the manuscript. We thank K. Matthews (Edinburgh) for providing PAD1 antibodies.

## Author Contributions

**Conceptualization:** Paula Ana Iribarren, Federico Rojas, Miguel Navarro, Vanina Eder Alvarez.

**Formal analysis:** Paula Ana Iribarren, Lucía Ayelén Di Marzio, Vanina Eder Alvarez.

**Funding acquisition:** Vanina Eder Alvarez.

**Investigation:** Paula Ana Iribarren, Lucía Ayelén Di Marzio, María Agustina Berazategui, Lorena Coria, Federico Rojas, Miguel Navarro.

**Methodology:** Paula Ana Iribarren, Federico Rojas, Miguel Navarro.

**Resources:** Federico Rojas, Miguel Navarro.

**Supervision:** Andreu Saura, Juliana Cassataro, Miguel Navarro, Vanina Eder Alvarez.

**Writing – original draft:** Vanina Eder Alvarez.

**Writing – review & editing:** Paula Ana Iribarren, Andreu Saura, Vanina Eder Alvarez.

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
