## [Decision Letter · Decision Letter 0]

19 Feb 2024

Dear Dr. Alvarez,

Thank you very much for submitting your manuscript "Depolymerization of SUMO chains induces slender to stumpy differentiation in T. brucei bloodstream parasites" for consideration at PLOS Pathogens. As with all papers reviewed by the journal, your manuscript was reviewed by members of the editorial board and by several independent reviewers. The reviewers appreciated the attention to an important topic. Based on the reviews, we are likely to accept this manuscript for publication, providing that you modify the manuscript according to the review recommendations.

The reviewers agree that that the phenotype of sumo chain elongation mutants in Trypanosoma is interesting and supported by the experimental evidence. Prior to acceptance, however, complete documentation of all experiments, including statistical support, must be added, and the data presentation and figures improved as indicated by the multiple requests and questions of reviewers.

Furthermore, the study suffers from a number of not logically imperative conclusions and massive over interpretation. Consequently, the text and discussion should be revised and specifically the following points should be taking into account:

1. There is no evidence that SUMOylation is involved in a physiological stumpy induction pathway or more generally in a signaling process. The mutant phenotype is clearly mimicking at least parts of the stumpy gene expression program, yet this does not mean that it is involved in SIF signaling.

2. It is well-known and published that physiological, and non-physiological stresses can induce the stumpy differentiation program, eventually by bypassing the physiological signaling process. The connection between SUMO and stress should be considered more carefully in this context (as an example Nat Rev Mol Cell Biol. 2022 Nov;23(11):715-731. doi: 10.1038/s41580-022-00500-y).

Conclusions and discussion should be adapted accordingly.

Sincerely,

Michael Boshart

Academic Editor

PLOS Pathogens

James Collins III

Section Editor

PLOS Pathogens

Michael Malim

Editor-in-Chief

PLOS Pathogens

orcid.org/0000-0002-7699-2064

The reviewers agree that that the phenotype of sumo chain elongation mutants in Trypanosoma is interesting and supported by the experimental evidence. Prior to acceptance, however, complete documentation of all experiments, including statistical support, must be added, and the data presentation and figures improved as indicated by the multiple requests and questions of reviewers.

Furthermore, the study suffers from a number of not logically imperative conclusions and massive over interpretation. Consequently, the text and discussion should be revised and specifically the following points should be taking into account:

1. There is no evidence that SUMOylation is involved in a physiological stumpy induction pathway or more generally in a signaling process. The mutant phenotype is clearly mimicking at least parts of the stumpy gene expression program, yet this does not mean that it is involved in SIF signaling.

2. It is well-known and published that physiological, and non-physiological stresses can induce the stumpy differentiation program, eventually by bypassing the physiological signaling process. The connection between SUMO and stress should be considered more carefully in this context (as an example Nat Rev Mol Cell Biol. 2022 Nov;23(11):715-731. doi: 10.1038/s41580-022-00500-y).

Conclusions and discussion should be adapted accordingly.

Reviewer Comments (if any, and for reference):

Reviewer's Responses to Questions

**Part I - Summary**

Reviewer #1: Well written and interesting piece of work. Major issues to do with data presentation and not conclusions per se. I would support publication after some textual/figure revision. Given that SUMO Is a known stress marker, I think it important to raise the possibility that the phenotype here represents a defective or modulated stress response/status and which may presage a stumpy-like state.

Reviewer #2: The authors generate bloodstream form T. brucei mutants unable to form SUMO chains on target proteins. In monomorphic forms this reduces virulence and renders parasites more sensitive to the CA procyclic differentiation signal. In pleomorphs, stumpy forms appear at lower parasitaemia and respond to CA procyclic differentiation at increased rates. These results are novel and highly interesting to the field as the mechanisms which regulate stumpy formation are of great importance for transmission of T. brucei. Furthermore, post-transcriptional mechanisms used by T. brucei for gene expression regulation are of particular interest across trypanosomes, due to the lack of transcriptional regulation. The finding that polySUMOylation impacts this step in the T. brucei life cycle may also lead to discoveries of its involvement at other stages, and in other related parasite species.

Overall, the experiments support the authors conclusions, however there are some gaps where the data are not shown or replicates are not clear.

Reviewer #3: This nice manuscript shows that bloodstream forms of African trypanosomes that are only able to conjugate SUMO monomers are primed for differentiation into stumpy transmissible forms. This was partially observed in monomorphic lines, that are theoretically unable to produce stumpy forms, in response to quorum sensing signaling in mice, and more unambiguously in pleomorphic parasites in which stumpy cells were observed at unusually low parasitemia levels. The authors propose that SUMO depolymerization may represent a coordinated signal triggered during stumpy activation program.

**Part II – Major Issues: Key Experiments Required for Acceptance**

Reviewer #1: None

Reviewer #2: (No Response)

Reviewer #3: The data are original and based on solid experimental evidence, yet the interpretation of the results, or at least the formulation of the conclusions, may be expressed in a more precise and cautious way along the entire manuscript. Taken together, these results suggest that, in contrast to monomorphic WT strains, the absence of polySUMOylation renders parasites more susceptible to differentiation, while host-parasite interaction is required to trigger stumpy-like cells. However, the authors claim, several times in the manuscript, that the absence of polySUMOylation can trigger this differentiation as a signal, which is not proven in this study.

Another important aspect is that monomorphic SUMO chain mutant parasites actually display limited stumpy-like characteristics. In fact, they show only a minimal part of the stumpy-like characteristics as (1) PAD1 proteins were not detected in monomorphic SUMO mutants, (2) monomorphic SUMO mutants were not cell-cycle arrested in vitro, (3) they could not be recognized as stumpy parasites based on their morphology, and (4) they still kill their host much more rapidly than a pleomorphic strain does. In total, only their transcriptome is indeed comparable, at least in part, to that of stumpy forms, which is already a very intriguing observation, and which supports well the authors’ hypothesis. This over-interpretation could be toned down along the entire manuscript.

**Part III – Minor Issues: Editorial and Data Presentation Modifications**

Reviewer #1: F1: Panel B - comment on residual possible SUMO adducts. Do the authors consider that these represent monoSUMOylated adducts? In panel D the authors should comment on the punctate staining and that the vast majority of the signal is likely to derive from unconjugated SUMO. I’m also not 100% convinced that the nuclear focus at panel D bottom might not be at the cytoplasmic face of the nucleus?

F2: Make clear if the graphs in panel B are representative examples or collated data?

F3: Panel A - WT seems to have very low G0 proportion; I tend to think of this as being closer to 80% in well growing cultures. I FOUND the ANNOTATION OF PANEL C SOMEWHAT UNCLEAR. FOR EXAMPLE THE TOP category was near invisible on my screen and I have no idea what Grumpy is. Overall I felt the data for this panel deserve some better discussion. I note that the data in Table S1 use old 427 annotations and it would be helpful if these were updated or included the 927 ortholog accession. Also, morphologically these do not look to be stumpy cells and is worth comment?

FS3: I’m not fully confident that the analysis here truly represents a test for the entire immune system. I think the point is well taken that the attenuation is not due to some massive immune response but could be diluted a little.

Reviewer #2: The data in figure 2b and S2 are unclear in the current format. The text states “In contrast, mice infected with SUMO chain mutants (n=17) showed two or three waves of parasitemia and in some cases even cleared the infection”

• How many mice cleared the infection? It is just one? The one shown in group 3?

• How many mice died as a result of the infection with the mutant line? Please indicate on the plots

• Was the experiment ended on day 11? Or did the mice die due to the infection at this point? Or did they clear the parasites at this point?

• It would be clearer to show the variation between mice in the main figure. Please add the data points/error bars to the trends in figure 2b.

In reference to monomorph infection, authors state “At the peak of parasitemia, we observed

cells that could not be recognized as genuine stumpy parasites based on their morphology

or biochemical markers (not shown) [19].” Please show these data, state the biochemical markers (PAD1?) and make clear the timepoint and mice these sample were taken from. It is necessary to show these data as it impacts the interesting finding that the loss of polySUMOylation increase the sensitivity of monomorphs to the CA signal without generating full stumpy forms.

Other points:

• In figure 1. Panel B and C are labelled incorrectly

• What is meant by “three-dimensional immunofluorescence” in reference to fig 1D? It’s not clear in the methods what 3D is referring too.

• In figure 3a, the plot shows “*” and legend says “**, -<0.01”, please clarify.

• In figure 3B, does M1-M5 indicate mouse 1- mouse 5?

• Please state the time points for the data in figure 3b and c. when was RNA collected?

• For figure 3D and E, please indicate replicates, exact counts and error bars.

• In figure 5A please add error bars

• For figure 5C, D and E, please indicate the time point the sample was made from, when in the peak? Were these taken on the same day for WT and mutants?

• In figure 5E, is this just one replicate? replicates should be shown.

• How were “intermediate cells” defined in 5E.

• The discussion states “In monomorphic parasites, mice infected with SUMO chain mutants showed relapsing and remitting waves of parasitemia, reaching densities not higher than 108/ml and clearing the day after.” Plots in 2B and supplementary show parasitaemia reaching 1x10^9/ml. It isn’t clear which mice cleared the parasites currently.

• Please deposit RNA-seq data to online repository and reference the accession number

Reviewer #3: Introduction

• Do Trypanosoma brucei spp parasites only live in the blood of infected animals?

Results

• ‘However, this difference does not affect VSG mRNA or protein levels (Figure S1), suggesting that mono- and/or multi-SUMOylation are sufficient to promote VSG expression.’ Do you mean that the absence of poly-SUMOylation does not affect VSG mRNA or protein levels?

• ‘At the peak of parasitemia, we observed cells that could not be recognized as genuine stumpy parasites based on their morphology or biochemical markers (not shown) [19]’ This is important and I think the reader would appreciate to see these cells.

• ‘In addition, PAD1 and 12 transcripts that are highly expressed in stumpy forms were also upregulated in these cells [20, 21].’ Were transcripts of the FHR and/or enzymes of the Ox-Phos pathway also upregulated in these cells?

• ‘Overall, these results suggest that the absence of polySUMOylation may stimulate the formation of stumpy-like cells, rendering monomorphic bloodstream parasites more sensitive to CA-triggered differentiation and that the underlying mechanism is upstream of AMP signalling.’ Please rephrase. These data do not show any stimulatory effects in absence of polySUMOylation.

• A bona fide comparison between different types of induction of ST differentiation would have more clearly assessed the position of the polySUMOylation effect along the differentiation pathway. For instance, a comparison between cAMP and spent medium or basement membrane extract medium.

• How could these monomorphic mutants differentiate into PCF upon CA induction without PAD1 and PAD2 detected at their surface?

• ‘As shown in Figure 4D and 4E, chain mutant parasites had a significantly higher proportion of cells expressing PAD1 (74% ± 2%), while only 35% ± 6% of WT cells showed expression of PAD1 at the same time point.’ It is not easy for me to detect any single PAD1+ cell in the picture presented in Fig4D for WT cells.

• Could the lower parasitemia observed during infections with the SUMO allKR strain in Fig 5B result from an increased level of parasite extravasation?

<b

---

## [Decision Letter · Decision Letter 1]

1 Apr 2024

Dear Dr. Alvarez,

We are pleased to inform you that your manuscript 'Depolymerization of SUMO chains induces slender to stumpy differentiation in T. brucei bloodstream parasites' has been provisionally accepted for publication in PLOS Pathogens.

Best regards,

Michael Boshart

Academic Editor

PLOS Pathogens

James Collins III

Section Editor

PLOS Pathogens

Michael Malim

Editor-in-Chief

PLOS Pathogens

orcid.org/0000-0002-7699-2064

the manuscript has been significantly improved during the revision and reviewers agree that their criticism and suggestions have been properly addressed.

Congatulations to this interesting report.

Reviewer Comments (if any, and for reference):

Reviewer's Responses to Questions

**Part I - Summary**

Reviewer #2: All my previous comments and questions have now been clearly answered by the authors.

Reviewer #3: (No Response)

**Part II – Major Issues: Key Experiments Required for Acceptance**

Reviewer #2: (No Response)

Reviewer #3: (No Response)

**Part III – Minor Issues: Editorial and Data Presentation Modifications**

Reviewer #2: (No Response)

Reviewer #3: (No Response)

PLOS authors have the option to publish the peer review history of their article (what does this mean?). If published, this will include your full peer review and any attached files.

Reviewer #2: No

Reviewer #3: **Yes: **Brice Rotureau

---

## [Editor Report · Acceptance letter]

15 Apr 2024

Dear Dr. Alvarez,

We are delighted to inform you that your manuscript, "Depolymerization of SUMO chains induces slender to stumpy differentiation in *T. brucei* bloodstream parasites," has been formally accepted for publication in PLOS Pathogens.

Best regards,

Michael Malim

Editor-in-Chief

PLOS Pathogens

orcid.org/0000-0002-7699-2064